# Nanosilver–Biopolymer–Silica Composites: Preparation, and Structural and Adsorption Analysis with Evaluation of Antimicrobial Properties

**DOI:** 10.3390/ijms252413548

**Published:** 2024-12-18

**Authors:** Magdalena Blachnio, Malgorzata Zienkiewicz-Strzalka, Jolanta Kutkowska, Anna Derylo-Marczewska

**Affiliations:** 1Department of Physical Chemistry, Institute of Chemical Sciences, Maria Curie-Sklodowska University, Maria Curie-Sklodowska Square 3, 20-031 Lublin, Poland; malgorzata.zienkiewicz-strzalka@mail.umcs.pl (M.Z.-S.); anna.derylo-marczewska@mail.umcs.pl (A.D.-M.); 2Department of Genetics and Microbiology, Institute of Biological Sciences, Maria Curie-Sklodowska University, 19 Akademicka Street, 20-033 Lublin, Poland; jolanta.kutkowska@mail.umcs.pl

**Keywords:** adsorption kinetics, adsorption equilibrium, dye adsorption, nanosilver–chitosan–silica composite, biopolymer, chitosan, silica

## Abstract

In this article, we report on the research on the synthesis of composites based on a porous, highly ordered silica material modified by a metallic nanophase and chitosan biofilm. Due to the ordered pore system of the SBA-15 silica, this material proved to be a good carrier for both the biologically active nanophase (highly dispersed silver nanoparticles, AgNPs) and the adsorption active phase (chitosan). The antimicrobial susceptibility was determined against Gram-positive *Staphylococcus aureus* ATCC 25923, Gram-negative bacterial strains (*Escherichia coli* ATCC 25922, *Klebsiella pneumoniae* ATCC 700603, and *Pseudomonas aeruginosa* ATCC 27853), and yeast *Candida albicans* ATCC 90028. The zones of microbial growth inhibition correlated with the content of silver nanoparticles deposited in the composites and were the largest for *C. albicans* (14–21 mm) and *S. aureus* (12–17 mm). The suitability of the composites for the purification of water and wastewater from anionic pollutants was evaluated based on kinetic and equilibrium adsorption studies for the dye Acid Red 88. The composite with the highest amount of the chitosan component showed the greatest adsorption capacity (a_m_) of 0.57 mmol/g and the most effective kinetics with a rate constant (log k) and half-time (t_0.5_) of −0.21 and 1.62 min, respectively. Due to their great practical importance, AgNP–chitosan–silica composites can aspire to be classified as functional materials combining the environmental problem with microbiological activity.

## 1. Introduction

Materials based on non-metallic structures are considered an innovative alternative to antibiotics, as they are effective against the phenomenon of the multidrug resistance of bacteria [1,2,3,4,5,6]. The combination of silver nanoparticles with biopolymer components gives hope for the creation of a promising agent against pathogenic microorganisms [7,8,9,10]. The combination of bactericidal silver nanoparticles with bacteriostatic chitosan increases the antibacterial activity of this biomaterial and may have a synergistic effect on all components [11,12,13,14]. Biopolymer nanocomposites as advanced and functional materials consist of nanoparticles dispersed in or coated with a biopolymer matrix [15,16,17,18,19]. During the synthesis process, the nanoparticles are tightly bound to chitosan by the chemical bonding between the electron-rich nitrogen atoms and the free orbitals of the silver ions [20,21]. The coating of silver nanoparticles with a polymer makes it possible to minimize their cytotoxic effect [22,23] and improve their biocompatibility. The specific morphology of the polymers and their chain structure enable the proper incorporation of nanoparticles (NPs) into the material network [24,25] and improve their biocompatibility. If the functional groups of polymers are selected in the right way, they take an active part in the synthesis by reducing metal ions or stabilizing the formed particles, which provides nanomaterials with long-term durability [26]. The effects of silver nanoparticles on bacterial cells are multidimensional (Figure 1), so that the mechanisms responsible for the biocidal activity of metal nanoparticles involve numerous cellular phenomena and processes [27,28,29,30]. They destroy the bacterial cell membrane (1) through the formation of reactive oxygen species (ROS), which leads to oxidative stress (2). They damage DNA by interacting with phosphate residues in the sugar–phosphate backbone (3) and impair replication and transcription processes and, ultimately, lead to DNA and RNA degradation (4). They impair the function of the respiratory chain, cell division, signal transduction systems, and the functions of numerous enzymes, especially those rich in sulfur-containing amino acids (5). They interfere with the production of adenosine triphosphate (ATP) and increase the susceptibility of proteins to denaturation (6). The presence of metallic nanoparticles can also cause uncontrolled ion transport and the leakage of metabolites and protons from inside the bacterial cell (7). Due to their ability to bind to nucleic acids, they also effectively increase the efficacy of currently marketed chemotherapeutic agents.

An important example of the microbiological activity of nanosilver forms is the influence of silver nanoparticles on increasing the activity of antibiotics against bacterial biofilms [31,32], which enhances the activity of antibiotics. Bacteria living in biofilms can easily spread throughout the body and lead to chronic and difficult-to-treat infections [33]. The biofilm environment creates favorable conditions for the growth of microorganisms. Microorganisms in the biofilm are characterized by a high resistance to biocidal agents, which also promotes the development of antibiotic resistance. The effectiveness of AgNPs in preventing the formation of biofilms and even in the removal (eradication) of formed biofilms on medical and industrial devices is currently being intensively investigated [34]. Silver nanoparticles not only have a high antibacterial potential, but also are able to interact with conventional antibiotics and, thus, enhance their effect [35,36,37]. Silver nanoparticles are becoming an important element in various aspects of biosecurity, including regular and ongoing disinfection in the veterinary field [38,39]. The undisputed usefulness of silver nanoparticles is also often confronted with environmental issues. Among the pollutants in aquatic ecosystems, besides microorganisms, there are often complex pollutants (such as dyes) that are not easily biodegradable [40,41]. Therefore, the elimination of biological hazards should be explored while adapting to the effective removal of pollutants other than just microbiological ones.

In all regions of the world, the high level of environmental pollution and the simultaneous emergence of new types of impurities and toxins are observed. Thus, studies on the development of new effective and selective adsorbents for the elimination of various types of pollutants from aquatic environments are being intensively elaborated by many research teams. The composite materials with silver nanoparticles proposed in this paper constitute an interesting proposal of adsorbents showing a high affinity to organic pollutants and simultaneous antibacterial properties. It is a composite material (chitosan–silica) that is structurally and physicochemically adapted to the adsorption of dye compounds and shows biological activity (due to the bound nanometallic phase). The obtained composites were characterized by applying various techniques to determine the textural, structural, and surface properties. Their antimicrobial properties were determined against selected bacterial strains, and the adsorption affinity towards selected dyes was studied.

## 2. Results and Discussion

### 2.1. Stability of Colloidal Silver Solution

In the synthesis of silver nanoparticles, stabilizers play a crucial role in preventing agglomeration. Such an effect can impair the unique properties of the nanoparticles, such as their high surface-to-volume ratio and biological activity. This is due to the high surface energy of metallic nanoparticles, which leads to their natural tendency to aggregate. To monitor the stability and kinetics of nanoparticle formation, UV–Vis spectroscopy was used to provide insights into the reduction process and the properties of the resulting colloidal solution. Silver nanoparticles exhibit a distinctive absorption peak in the UV–Vis spectrum due to the surface plasmon resonance (SPR) effect, in which collective oscillations of conduction electrons occur in response to incident light. This SPR signal typically appears in the wavelength range of 200–800 nm. Figure 2A shows the absorption peak at around 450 nm, confirming the successful synthesis of silver nanoparticles. The UV–Vis measurements were performed at 5 min intervals and show that the reduction of diamminesilver(I) ions was completed within 90 min. After this time (Figure 2B), the nanoparticles remained stable for longer periods of time. No significant changes in the intensity or position of the absorption peak were observed after 1.5 h, 2 h, 3 h, 24 h, and even up to one and four weeks.

The lack of peak shifts or intensity changes over time indicates that the stabilizer effectively prevents agglomeration and ensures that the nanoparticles maintain their size, distribution, and functional properties over time. This stability is crucial for maintaining the desired properties of the nanoparticles in applications such as catalysis, sensing, and biomedicine.

### 2.2. Assessment of the Presence of Silver Nanoparticles in Composite Materials

To confirm the presence of silver nanoparticles in the obtained composites, UV–Vis measurements were performed using a solid-state device. In general, for metallic silver with a high dispersion, a peak related to the plasmon effect should be visible in the spectrum. This effect is not generated by derivatives of the silver phase such as silver chloride and silver oxide. Using the surface-localized plasmon effect, it is, therefore, possible to confirm the presence or absence of silver nanoparticles in a given material. Figure 3 shows the UV–Vis spectra of pure SBA-15 and composites containing silver nanoparticles (AgNPs).

In the case of the pure silica sample, the obvious result is the absence of an absorption signal in the spectrum. For the other samples, a clear signal is visible at a wavelength of 400 nm, confirming the presence of silver nanoparticles. The signal intensity increases with the increase in their concentration in the analyzed samples. It should also be noted that the absorbance signal is shifted to lower wavelengths compared to the position of this signal obtained for the initial silver nanoparticle solution. This could be related to the reduction in size of the nanoparticles due to immobilization on the solid surface.

### 2.3. Characterization of the Composite Materials

Nitrogen adsorption analysis was performed to determine the textural properties of the composite materials. The nitrogen adsorption/desorption isotherms for the samples AgChS1, AgChS2, and AgChS3 are shown in Figure 4A. All materials show a type IV isotherm with an H1 hysteresis loop, which is typical for well-organized structures with uniform interconnected mesopores. The profile corresponds to monolayer adsorption up to the first inflection point of the curve (relative pressure (p/p_0_) of ~0.1) and changes to multilayer adsorption above this value. Then, hysteresis occurs in the range of p/p_0_ from 0.42 to 0.82. The sample AgChS1 is characterized by a large specific surface area of 527 m^2^/g (Table 1). The surface area decreased to 444 m^2^/g (AgChS2) and 396 m^2^/g (AgChS3) as a result of introducing a larger number of modifying compounds. The same decreasing trend is observed for the pore volume. The total pore volume, V_t_, diminishes from 0.64 cm^3^/g (AgChS1), to 0.54 cm^3^/g (AgChS2), to 0.48 cm^3^/g (AgChS3). The calculated values of the primary mesopores volumes, V_p_, show a similar behavior: from 0.52 cm^3^/g (AgChS1), to 0.45 cm^3^/g (AgChS2), to 0.40 cm^3^/g (AgChS3). It should also be underlined that mesopores constitute the majority of the porous structure of synthesized materials.

An analysis of the pore size distribution from the adsorption branch of the isotherms (Figure 4B) shows the presence of mesopores of sizes (determined from the peak maxima) 6 nm, 7.5 nm, and 8 nm for the samples AgChS3, AgChS2, and AgChS1, respectively.

The pore size distributions calculated from the desorption branches show slightly smaller values, namely, 5.5 nm, 6 nm, and 6.5 nm for the samples AgChS3, AgChS2, and AgChS1 (Figure 4C). In this case, a peak for pores of size 3.4 nm is also visible, which is related to the cavitation pressure. The differences in pore size are, therefore, related to the number of nanosilver and chitosan phases that are part of the material. The pore size decreases in the order AgChS1 > AgChS2 > AgChS3, which is explained by the increasing number of modifiers in the composites (AgChS1 < AgChS2 < AgChS3). A similar trend, although with slightly different pore size values, shows the BJH calculations of pore sizes in the tested materials (Table 1).

The acid–base properties of AgNP–chitosan–silica composites were investigated by potentiometric titration. The recorded data were used to determine the dependence of the surface charge density on the solution pH (Figure 5) and the value of the zero charge point (pH_PZC_) for the tested solids. By definition, the pH_PZC_ corresponds to the pH value of the solution at which the surface charge of the solid is zero. According to the literature, the pH_PZC_ for pure SBA-15 silica is in the range of 4.5–5 [42,43], while, for the tested composites AgChS1, AgChS2, and AgChS3, it is 6.1, 6.3, and 6.5, respectively. The modification of silica by the immobilization of nanoparticles and biopolymer on its surface led to a change in the chemical nature of the material to more basic, which is reflected in the shift of the considered parameter towards higher values. The more AgNPs and chitosan are used in the synthesis, the more positive a charge can be generated on the surface of the obtained material. Since the biopolymer phase predominates in the tested materials, the observed changes in the electrochemical character of the composites are mainly determined by this phase.

Figure 6 shows the small-angle X-ray diffraction patterns of the investigated materials. The XRD patterns of the materials show three reflections at 2θ values between 0.5– and 2.0, including a strong peak (100) at 0.89, 0.92, and 0.84 degrees 2θ for the (100) peaks of the AgChS1–AgChS3 samples. The closest peaks are at 1.63, 1.62, and 1.77 deg 2θ for the (110) plane and at 1.88 and 1.85 deg 2θ for (200). This corresponds to the preservation of the structure corresponding to the ordered hexagonal mesoporous silica framework and a two-dimensional (2D) hexagonal mesostructure with the space group *p6mm* [44]. The XRD results showed that the ordered structure of the silica components deteriorated with the addition of the silver phase. Reaching the peak (100) in the XRD curves is an indication of the mesoporous structure. However, the decrease in all peak intensities, the line width, and even the disappearance of the (200) reflectance for the AgChS3 material indicate a gradual blurring of the ordering structure across the entire composite structure due to the presence of the biopolymer phase and the nanometallic inclusions. However, if you focus only on the dimension of the lattice constant of the hexagonal phase for individual samples (*a_av_.* = 11 nm, 10.9 nm, and 11 nm for AgChS1, AgChS2, and AgChS3, respectively), similar values remain, which could indicate a slight influence of the composite components in the microstructure of the silica phase itself.

The XRD patterns of the biopolymer–nanosilver composites in Figure 7 show a conspicuous broad peak between 2θ = 15° and 30°, indicating the presence of amorphous silica particles (10.5772/62033). This signal was also confirmed for the silica reference (orange XRD curve). In contrast, the AgChS1–AgChS3 composites show five distinct diffraction peaks at 38°, 44°, 64°, 77°, and 82°, corresponding to the (111), (200), (220), (311), and (222) planes of the face-centered cubic silver nanoparticles as identified by JCPDS map number 04-0783.

The full width at half maximum (HWHM) values of the Ag peaks were used to calculate the average size of the silver crystallites in all visible lattices (Figure 8). The calculations showed that the size of the silver particles considering their crystallinity area was 22.5 nm, 23.4 nm, and 23.8 nm for samples AgChS1, AgChS2, and AgChS3, respectively. Similar values of silver crystallite sizes for samples prepared with increasing amounts of silver nanoparticles indicate a good stabilization of the metallic phase at the surface and a low degree of agglomeration, which could affect the biological activity of the tested materials. The biopolymer component was only detected in the case of the AgChS3 sample, where the addition of the biopolymer phase during synthesis was the largest. This sample shows an amorphous signal in the initial part of the XRD curve at the 10-degree position. The phase purity of the materials obtained after synthesis is also remarkable, as only metallic silver is visible.

The TEM images of the AgNP–chitosan–silica composites (Figure 9) show the existence of a highly uniform and ordered hexagonal pore arrangement of the silica phase (SBA-15), which is maintained even after the introduction of the AgNP component and functionalization by the chitosan phase. This structural organization is also confirmed by the small-angle X-ray scattering pattern shown in Figure 6. The biopolymer component forms a thin layer on the surface of the substrate, partially covering the silica phase and the silver nanoparticles. However, on large areas of the sample, the metallic nanoparticles remain accessible to external media, which is why the antibacterial properties confirmed by the microbiological tests presented below are maintained. The size of the silver nanoparticles is in the range of 20–30 nm. When analyzing the TEM images, an increase in the amount of silver nanoparticles was also observed for the AgChS3 sample compared to the sample with the lowest content of this component (AgChS1), which is consistent with expectations and the XRD results (Figure 7). The TEM image of hte initial AgNP and the TEM image with the EDX mapping of the AgChS3 sample (as a control) were presented in the Appendix A, respectively.

### 2.4. Microbiological Activity of AgNP–Chitosan–Silica Composites

In this study, the antimicrobial activity of AgNP–chitosan–silica composites (AgChS1–AgChS3), AgNPs (initial silver nanoparticles), and chitosan–silica composites (ChSBA, a material without a metallic nanophase) were tested against Gram-positive *Staphylococcus aureus* ATCC 25923, Gram-negative bacterial strains (*Escherichia coli* ATCC 25922, *Klebsiella pneumoniae* ATCC 700603, and *Pseudomonas aeruginosa* ATCC 27853), and the yeast *Candida albicans* ATCC 90028. All tested AgNP–chitosan–silica composites showed antimicrobial activity, and the size of the growth inhibition zones correlated with the silver nanoparticle content. The largest zones of growth inhibition were observed for *C. albicans* (14–21 mm), followed by *S. aureus* (12–17 mm). For Gram-negative bacteria, the zones of inhibition were smaller for the drug-resistant strains *K. pneumoniae* and *P. aeruginosa* (10–12 mm and 10–14 mm, respectively) and for the antibiotic-susceptible strain *E. coli* (11–14 mm). The composite AgChS3 with the highest AgNP content showed the highest activity with growth inhibition zones of 12 to 14 mm for Gram-negative strains, 17 mm for *S. aureus*, and 21 mm for *C. albicans* (Table 2, Figure 10).

The chitosan–silica composite ChSBA showed no antibacterial or antifungal activity, AgNPs showed moderate activity against bacterial strains (growth inhibition zone 7–9 mm), and the best activity was observed against *C. albicans* (11 mm). The antimicrobial activity of the tested materials was compared with the standard antibiotic ampicillin for bacterial strains and amphotericin B for yeasts. The growth inhibition zones for ampicillin were 30 mm and 20 mm for *S. aureus* and *E. coli*, respectively. The remaining two strains were resistant to ampicillin. The *C. albicans* strain tested was sensitive to amphotericin B with an inhibition zone of 21 mm. Based on the studies performed, it can be confirmed that chitosan increases the activity of AgNPs; the tested composites showed promising antifungal and also antibacterial effects, especially against *S. aureus*.

To visualize the biological activity of the tested composites on selected bacterial strains, an AFM analysis of Gram-positive *Staphylococcus aureus* ATCC 25923 and Gram-negative *Escherichia coli* ATCC 25922 was performed before (control sample) and after contact with the composite material (Figure 11 and Figure 12 for *S. aureus* and *E. coli*, respectively). The AFM analysis showed well-developed morphological forms of the bacteria with a shape and size suitable for the respective strains. *Staphylococcus* cells are round, large (1 μm), convex, and opaque. In the case of *E. coli*, a rod-shaped morphology was confirmed and a unit cell size of approximately 3 μm in length and 1 μm in width. AFM analysis allowed the visualization of changes that occurred in the bacterial systems as a result of contact with the composite material (AgChS3) after a relatively short contact time (18 h). In both cases, changes were observed in the outer layer of polysaccharides covering the cells (bacterial capsule). The well-developed and visible shape of the capsule in the control samples (Figure 11A,C,E and Figure 12A,C,E) disappears almost completely in the samples exposed to the composite material with silver nanoparticles (Figure 11B,D,F and Figure 12B,D,F). In the case of *S. aureus*, this effect is even more pronounced. The environment of the bacteria is distorted, with visible depressions, irregularities, and bulges (Figure 11D,F). In the case of *E. coli*, the envelope disappears and accumulated nanostructures of foreign material and localized perforations become visible (Figure 12E,F). In both cases, the bacterial surface was rougher with spherical nanometallic shapes located within the surface layer. The results obtained are consistent with the mechanisms of bactericidal action of materials containing metallic nanoparticles, with the first phase indicating an attack on the cell shields and defense functions of the bacterial cells.

Several mechanisms have been described to explain the antibacterial activity of AgNPs in composite systems. The most common are the gradual release of free silver ions and the subsequent disruption of ATP production and DNA replication, damage to proteins in the bacterial cell wall, and the generation of reactive oxygen species, which can ultimately lead to cell death. Many authors have reported the antimicrobial effect of chitosan on bacteria and fungi. This activity is influenced by both the physicochemical properties of chitosan and the nature of the target microorganism. High-molecular-weight chitosan does not penetrate the bacterial cell, acts as a metal chelator, prevents the passage of nutrients out of the cells, and alters cell permeability. Low-molecular-weight chitosan acts intracellularly, affecting RNA, protein synthesis, and mitochondrial function [45]. The combination of chitosan (N-alkylaminated nanoparticles) with antibiotics has been shown to significantly reduce the minimum inhibitory concentration (MIC) values for multidrug-resistant *E. coli* strains that produce β-lactamases, for ampicillin and piperacillin (10,000- and 1000-fold), and for cefoxitin and ceftazidime (100- and 20-fold) [46]. Chitosan increases the activity of AgNP complexes and can act as both a stabilizer and a carrier. In addition, the increase in surface area and charge enables better interaction with the negatively charged bacterial cell membranes, leading to a change in permeability and better penetration of the nanoparticles into the bacterial cell [47,48]. The proposed materials could be used in food preservation as antimicrobial agents and to extend shelf life. The in vivo antibacterial activity against *E. coli* in minced meat samples was confirmed, and the effect of Ch–AgNP was concentration-dependent [49].

### 2.5. Anionic Dye Adsorption on Composites

The presence of the chitosan component gives reason to believe that the tested materials will prove useful as adsorbents for the purification of water and wastewater from anionic organic pollutants. Chitosan, a product resulting from the deacetylation of chitin (waste product from seafood processing), has a polycationic character resulting from the numerous amine groups in its macromolecule. The formation of a composite structure consisting of (i) a silica matrix doped with silver nanoparticles and (ii) a biopolymer is based on the mechanism of interaction between a part of the amine/amide groups derived from chitosan and the silanol groups of the silica phase. The remaining free amine/amide groups may represent potential active adsorption sites for substances with the appropriate affinity. To evaluate the effectiveness of the composites AgChS1, AgChS2, and AgChS3 in a role of adsorbents, the anionic dye Acid Red 88 (AR 88), which belongs to the group of sulfonated azo compounds, was used. The physicochemical properties of the dye are as follows: molecular weight = 400.4 g/mol; acidity constant = 11; water solubility = 1.5 g/L; and distance between the most remote atoms in the dye molecule = 1.4 nm. The adsorption studies included both kinetics and equilibrium adsorption. The results of the latter are shown in Figure 13A in the form of experimental adsorption isotherms and fitting curves by the Generalized Langmuir (GL) isotherm equation [50,51,52,53]:(1)$$\theta ={\left[\frac{{\left(Kc_{eq}\right)}^{n}}{1+{\left(Kc_{eq}\right)}^{n}}\right]}^{m/n}$$
where θ is the global adsorption isotherm, c_eq_ is the equilibrium concentration (mmol/L), m and n are the heterogeneity parameters characterizing a shape (width and asymmetry) of the quasi-Gaussian adsorption energy distribution function, and K is the adsorption equilibrium constant.

The values of the parameters m and n can be within the limits (0; 1>) and, depending on the configuration adopted by them, Equation (1) is simplified to four simpler equations: Langmuir (L) (GL: m = n = 1), Langmuir–Freundlich (LF) (GL: 0 < m = n ≤ 1), Generalized Freundlich (GF) (GL: n = 1, 0 < m ≤ 1), and Tóth (T) (GL: m = 1, 0 < n ≤ 1). The fitting curves shown in Figure 13A correspond to the Generalized Freundlich (GF) equation. The values of the parameters of the GF isotherm equation for the tested adsorption systems, i.e., the adsorption capacity (a_m_), the heterogeneity parameters (m, n), and the logarithm of the adsorption equilibrium constant (log K), are listed in Table 3. Analyzing the course of the isotherms, systematic changes correlated well with the composition of the adsorbents can be observed. The adsorption capacity a_m_ increases with the amount of the biopolymer component in the composite and is 0.40, 0.49, and 0.57 mmol/g for AgChS1, AgChS2, and AgChS3, respectively. This is confirmed by the linear relationship of the adsorption capacity as a function of the nitrogen content in the sample, a_m_ = f(%N) (Figure 13B). The nitrogen content in each composite was determined by elemental analysis, and is 0.72%, 0.94%, and 1.19% for AgChS1, AgChS2, and AgChS3, respectively (Table 4). However, there is no correlation between the adsorption capacity and the porosity of the composites. This means that the number of amine or amide groups present on the surface of the composites is a key factor in determining the efficiency of the removal of the anionic dye from the solution. The increase in the a_m_ values is accompanied by a simultaneous decrease in the log K values, which are as follows: 0.04, −0.19, and −0.34 for AgChS1, AgChS2, and AgChS3, respectively. This is due to the presence of the metallic nanophase in the materials, which weakens the strength of the adsorbate–adsorbent interactions and their mutual affinity. In the inset of Figure 13A, the adsorption isotherm of Acid Red 88 on the chitosan–silica composite ChSBA as a control material is also presented. It was prepared using the same procedure and parameters as the composite AgChS3, only without the step of doping with AgNPs. As previously reported [54], the values of a_m_ and log K for ChSBA are equal to 0.78 mmol/g and 2.60, respectively. Taking into account the results cited and those presented here, it can be concluded that increasing the amount of silver nanoparticles in the composites leads to a decrease in the values of a_m_ and log K, which corresponds to a weakening of the affinity between the adsorbate and adsorbent. Despite the lower efficiency of AgNP–chitosan–silica composites compared to the control sample, their dual functionality as antimicrobial and environmental cleansing materials should be emphasized.

Analyzing the values of the parameter m of the GF isotherm equation for individual adsorption systems, it can be seen that the increase in the amount of metallic nanophase (as well as the content of the chitosan component) increases the energetic heterogeneity of the composites with respect to the adsorbate. For AgChS3, the value of the parameter m is the lowest, which means the greatest heterogeneity of the material. The adsorption mechanism of the dye Acid Red 88 on the AgNP–chitosan–silica composites is based on electrostatic interactions (dye anions–positive amine groups of the chitosan component) and hydrogen ones (amine, amide, hydroxyl, and azo groups of the dye–sulfonic, and hydroxyl groups of the composite).

Figure 14 shows the adsorption kinetics of AR 88 on the AgNP–chitosan–silica composites as profiles of the changes in relative concentration as a function of time (Figure 14A,C) and relative concentration as a function of the square root of time (Figure 14B,D).

Generally, the adsorption process is very fast (straight line section of the square root function of time) and then gradually slows down, although only the use of the composite AgChS3 allows the complete removal of the pollutant from the solution. When using the other adsorbents, part of the adsorbate remains in the unbound form in the solution at the end of the experiment. The differentiation of the efficiency of the composites is consistent with the results of the equilibrium adsorption studies, according to which the adsorption capacity correlates with the amount of chitosan component in the material and decreases as follows: AgChS3 > AgChS2 > AgChS1.

Many kinetic models and equations were used to optimize the kinetic data, i.e., the intra-particle diffusion model (IDM), pore diffusion model (PDM), first-order equation (FOE), second-order equation (SOE), and mixed 1.2-order equation (MOE), but the best results were obtained with the multi-exponential equation (m-exp) (Equation (2)) and the fractal-like MOE equation (f-MOE) (Equation (3)) [55,56,57,58]:(2)$$c=\left(c_{0}-c_{eq}\right)\sum_{i=1}^{n}f_{i}exp\left(-k_{i}t\right)+c_{eq}$$
where c is the temporary concentration, the c_0_ and c_eq_ are the initial and equilibrium concentrations, respectively, “i” is the term of the m-exp equation, and k_i_ is the rate coefficient.
(3)$$F=\frac{{1-exp\left(-k_{1}t\right)}^{p}}{{1-f_{2}exp\left(-k_{1}t\right)}^{p}}$$
where k_1_ is the rate coefficient, F is the adsorption progress, and p is the fractal parameter.

The multi-exponential Equation (2) describes adsorption in the so-called compartment model with a series of first-order processes that run in parallel and/or in sequence. It generally provides good results even if the process takes place on energetically or structurally heterogeneous solids and cannot be represented by simple kinetic equations, i.e., FOE, SOE, or MOE. In the procedure of optimizing kinetic data using the multi-exponential equation, it is possible to choose the number of exponential terms. In the analysis of the kinetic data, the number of exponents (n) was set to 1, 2, and 3. For n = 1, the m-exp equation is equivalent to the first-order FOE equation. For each variant of the equation, the relative standard deviation SD(c)/c_0_ was determined, on the basis of which the most optimal variant was specified. For two adsorption systems (dye/AgChS1 and dye/AgChS2), a subsequent increase in the number of exponential terms significantly improves the optimization quality. Finally, three exponents seem to be sufficient to describe their kinetics. For the dye/AgChS3 system, the differences in the SD(c)/c_0_ values for the considered variants of the m-exp equation (1-exp, 2-exp, and 3-exp) are not so significant, and, in the case of the latter two, they are comparable. This means that, among the tested AgNP–chitosan–silica composites, AgChS3 is characterized by the lowest structural heterogeneity. The dependence of the standard deviations of the relative concentration SD(c)/c_0_ on the number of exponential terms in the multi-exponential equation are presented in Figure 15A.

The fractal-like MOE equation (f-MOE) (3) is derived from the MOE equation and takes the fractality of the objects into account. In the optimization procedure, an additional parameter (p-fractal coefficient) is introduced, which is related to the non-ideality of the system. Similarly to m-exp, the f-MOE equation describes the adsorption based on a certain distribution of rate coefficients, not by specific numerical values of k_i_, but by the value of the parameter p. The more the fractal parameter deviates from the value 1 (typical for the non-fractal MOE equation), the larger the distribution of the rate coefficients can be expected. Therefore, the f-MOE equation is suitable for the description of effects far from ideality. In the optimization process using the f-MOE equation, values of the parameter f_2_ equal to or close to 1 were obtained for all systems, so that f-MOE (for f_2_ ∈ <0, 1>) was reduced to f-SOE. The theoretical lines and the analysis of the fit to the individual experimental points using the m-exp and f-MOE equations are presented in Figure 14A,B and Figure 14C,D, respectively. The parameters of the considered kinetic equations and the quantities used in the evaluation of the accuracy of the description of the adsorption kinetics are listed in Table 5.

The analysis of the kinetic parameters, i.e., the half-time (t_0.5_), the logarithm of the rate constant (log k), and the times needed to obtain the set decolorization efficiency (percentage of dye removal), shows a very rapid progress of dye adsorption on the composites. The values of t_0.5_ and log k (based on the 3-exp equation) are in the range of 1.58–2.12 min and −0.49–−0.36, respectively. The kinetic parameters of the studied systems are thus comparable and their values indicate an easy accessibility of the active adsorption sites for the dye molecules. This is undoubtedly related to the mesoporous structure of the composites, which is characterized by a developed surface area S_BET_ = 396–527 m^2^/g and a large total pore volume V_t_ = 0.48–0.64 cm^3^/g. The size of the dominant pores, read from the maximum of the pore size distribution (PSD) curves, is 6–8 nm (from the adsorption branch) and 3 nm and 6 nm (from the desorption branch). It follows that the structural parameters favor rapid adsorption. On the other hand, the analysis of the data in relation to the efficiencies obtained, i.e., 50%, 60%, 70%, and 80%, shows that the critical factor limiting the kinetics for the adsorption systems tested is the chitosan content in the composite (Table 6).

The composite AgChS3 with the highest content of the biopolymer component is characterized by the shortest times needed to achieve the set decolorization efficiency. Although this material is characterized by the least developed porosity, the situation is the opposite for the composite AgChS1; the lowest content of biopolymer in the material determines a relatively high-porosity structure but limits the number of active adsorption sites. Ultimately, for all three composites, the times corresponding to the efficiency of 60% and above begin to gradually differentiate. The higher the adsorption efficiency analyzed, the greater the differences in time visible. To achieve an 80% efficiency, the dye adsorption on AgChS3, AgChS2, and AgChS1 takes 6.75, 17.50, and 32.79 min, respectively, which corresponds to an approximately 5-fold difference in the process speed on the first and last material. The relationship between the adsorption kinetics at fixed values of the process progress and the content of the chitosan component in the composites (expressed as a percentage of N, which is 0.72%, 0.94%, and 1.19% for AgChS1, AgChS2, and AgChS3, respectively) is shown in Figure 15B. For all four analyzed efficiencies, linear dependencies with a high correlation coefficient R^2^ were obtained, which confirms the validity of the thesis that the chitosan content in the composite is a factor determining the adsorption rate.

In the data optimization procedure using the m-exp equation, the adsorption half-time parameter (t_0.5_) is the average value calculated numerically from the adsorption half-times t_0.5i_ for the respective equation terms (t_0.5i_ = (ln 2)/k_i_). Figure 15C,D presents the distribution of t_0.5i_ and log k_i_ as the relative contribution f_i_ of the kinetic terms in the m-exp equation. The narrower the distribution of the parameters, the smaller the differentiation of the rates in the successive stages of the process. Faster kinetics, on the other hand, can be recognized by the larger contribution of shorter half-times t_0.5i_ and higher rate constants k_i_. All three spectra have a similar shape and position in the coordinate system. However, on closer inspection, it can be seen that the spectra for AgChS2 and AgChS3 are slightly shifted towards the parameter values indicating a faster adsorption progression compared to the spectrum for AgChS1. Moreover, the spectrum for the composite with the highest chitosan content covers the narrowest range of t_0.5i_/log k_i_; i.e., it corresponds to the least differentiated adsorption rate. It should be remembered that f_i_ on the ordinate axis is a relative value and refers only to the adsorbate that was bound on the adsorbent surface. It does not take into account the amount of adsorbate remaining in the solution at the end of the experiment. For the systems with the composites AgChS1 and AgChS2, it is 9% and 4%, respectively, relative to the initial amount of adsorbate. Therefore, the presented relationships do not fully reflect the actual behavior of adsorption systems characterized by such differentiated efficiency.

Satisfactory results in the optimization of the kinetic data were obtained by using the fractal-like MOE equation (f-MOE) and, more precisely, its reduced form, i.e., f-SOE. However, based on the values of the coefficient of determination 1-R^2^ and the relative standard deviation SD(c)/c_0_, it appears that the optimization method using the f-SOE equation is less flexible than using the multiparameter m-exp equation. Nevertheless, the main kinetic parameters, such as the logarithm of the rate constant (log k), the adsorption half-time t_0.5_, and the entire adsorbate uptake (u_eq_), obtained using both methods are similar. In the f-MOE equation and its equivalent forms, a fractal parameter p was introduced to describe the kinetics with a certain distribution of rate coefficients. The values of this parameter for the adsorption systems with AgChS1, AgChS2, and AgChS3 are 0.70, 0.71, and 0.80, respectively. Lower p values for the first two systems indicate a larger distribution of rate coefficients compared to the last system. These results are consistent with the distributions of log k_i_ determined based on the m-exp equation (Figure 15D).

## 3. Conclusions

In this study, the synthesis of composites using a porous, highly ordered silica material modified with a metallic nanophase and a chitosan biofilm is presented. The silica SBA-15 demonstrates its effectiveness as a carrier for both the biologically active nanophase and the adsorption active phase.

The composites exhibit varying textural, structural, and surface properties based on the concentrations of modifying compounds used. Higher concentrations of modifiers result in a reduction in porosity and an enhancement in the basicity of the material surface.

All AgNP–chitosan–silica composites show antimicrobial activity, with the size of the growth inhibition zones correlating with the silver nanoparticle content. For the AgChS3 composite, growth inhibition zones measure 12 to 14 mm for Gram-negative strains, 17 mm for *Staphylococcus aureus*, and 21 mm for *Candida albicans*. The primary mechanism of antibacterial activity is identified as the gradual release of free silver ions, which leads to cell death.

The concentration of amine or amide groups on the surface of the composites is a key factor influencing the efficiency of the anionic dye removal from the solution. The adsorption capacities for AR 88 are as follows: 0.40, 0.49, and 0.57 mmol/g for AgChS1, AgChS2, and AgChS3, respectively. Notably, the increase in adsorption capacity is accompanied by a decrease in the log K value. This trend is attributed to the presence of the metallic nanophase in the materials, which weakens the interactions between the adsorbate and adsorbent, reducing their mutual affinity.

Kinetic studies reveal that dye adsorption on the AgNP–chitosan–silica composites occurs rapidly, with half-life values (t_0.5_) ranging from 1.58 to 2.12 min and log k values between −0.49 and −0.36. This suggests that the active adsorption sites are easily accessible to the dye molecules, which is attributed to the mesoporous structure of the composites. The composite AgChS3, containing the highest content of the biopolymer component, achieves the fixed decolorization efficiencies in the shortest times.

The adsorption mechanism of the dye Acid Red 88 on the AgNP–chitosan–silica composites involves two main interactions: (i) electrostatic interactions between the negatively charged dye anions and the positively charged amine groups of the chitosan component; and (ii) hydrogen bonding between the amine, amide, hydroxyl, and azo groups of the dye and the sulfonic and hydroxyl groups of the composite.

## 4. Materials and Methods

### 4.1. Materials and Chemicals

Pluronic P123 copolymer (average molecular weight of ~5800 Da), trimethyloctadecylammonium bromide (C_18_TAB, 98% purity), tetraethyl orthosilicate (TEOS, ≥99% purity), silver nitrate (AgNO_3_, >99% purity), sodium hydroxide (NaOH, >97% purity), polyvinylpyrrolidone (PVP, average molecular weight of ~29,000 Da), chitosan from shrimp shells (deacetylation degree of ~75%, molecular weight of ~190,000–370,000 Da), and Acid Red 88 (dye content of 75%) were purchased from Sigma-Aldrich (Poznan, Poland). Ammonia water (NH_4_OH, concentration of 25%), hydrochloric acid (HCl, concentration of 35–38%), acetic acid (CH_3_COOH, concentration of 99%), and formaldehyde (HCHO, concentration of 36–38%) were supplied by Polish Chemical Reagents (Avantor Performance Materials Poland S.A., Gliwice, Poland).

### 4.2. Synthesis of Silver Nanoparticle–Silica Phases

The synthesis of SBA-15-type silica was carried out according to the procedure described in [54]. The synthesis of the colloidal solution of silver nanoparticles, AgNPs, was carried out as follows: 20 mL of a 5% silver nitrate solution was added to 20 mL of a 5% sodium hydroxide solution. The resulting brown precipitate of silver oxide (Ag_2_O) was separated from the solution and washed with distilled water. As a result of the complexation reaction of silver oxide with ammonia, a colorless, water-soluble silver diammonium (I) hydroxide ([Ag(NH_3_)_2_]OH) was obtained. Then, 0.25 mL of silver diammonium (I) hydroxide was added to 50 mL of a 1% P123 copolymer solution and stirred with a magnetic stirrer for about 5 min. After this time, eight drops of formaldehyde were added to reduce silver diammonium ions to the metallic form. The reduction of Ag^+^ ions with formaldehyde was gradual, as evidenced by the change in the color of the solution from colorless to green–yellow, which is the characteristic color of a colloidal solution of small silver nanoparticles. The scheme of the synthesis process of silver nanoparticles, AgNPs, is shown in Figure 16.

In the next step, the obtained silver nanoparticles were deposited on silica material. For this purpose, 2 g of SBA-15 silica samples were placed into three Erlenmeyer flasks, and appropriate amounts of silver nanoparticle solution and distilled water (Table 4) were added. The reaction mixtures were placed in a thermostatic shaker (25 °C, 120 rpm) for two days. After this time, the AgNP–silica phase was decanted, washed with distilled water, and dried at 60 °C.

### 4.3. Synthesis of Composites Based on Chitosan and AgNP–Silica Phase

AgNP–chitosan–silica composites were obtained by impregnating the AgNP–silica phase with dissolved chitosan. In brief, the AgNP–silica phase (2 g) was added to three Erlenmeyer flasks containing 30 mL of the corresponding chitosan solution dissolved in dilute acetic acid (the amounts of chitosan used to prepare the solutions are given in Table 4). The mixtures were placed in a water bath (40 °C) with stirring for two days and then transferred to a dryer (60 °C) until the water was completely evaporated. The obtained AgNP–chitosan–silica composites were washed with distilled water, dried again, and ground in a mortar. During the synthesis of the composites, the amounts of selected reagents (AgNPs and chitosan) were changed, resulting in materials with different elemental composition and structural properties. A higher number in the name of the composite sample means a higher amount of AgNPs and chitosan used for the synthesis.

### 4.4. Microbiological Study

The antimicrobial properties of the AgNP–chitosan–silica composites were determined using the agar well diffusion method against bacterial strains: *Staphylococcus aureus* ATCC 25923, *Escherichia coli* ATCC 25922, *Klebsiella pneumoniae* ATCC 700603, *Pseudomonas aeruginosa* ATCC 27853, and the pathogenic yeast *Candida albicans* ATCC 90028 from the American Type Culture Collection (ATCC). Tested strains were cultivated on Muller–Hinton (MH) broth in 37 °C for 16 h and diluted in fresh medium to the density of 1.5 × 10^8^ CFU/mL (0.5 McFarland). The bacterial suspensions were then mixed with 3 mL of soft MH agar (0.7%) and poured onto MH agar plates. Holes with a diameter of 6 mm were drilled into the inoculated plates using a sterile cork borer, and 10 mg of the tested compounds (AgChS1-AgChS3) were added to the wells. Chitosan–silica composite ChSBA (10 mg) and 50 μL of silver nanoparticle solutions AgNPs (concentration of ~40 mg/L by atomic absorption spectrometry) were used as a control. The zones of bacterial growth inhibition were measured after 18 h of incubation at 37 °C. In addition, the sensitivity of the tested bacterial strains to ampicillin (BioMaxima S.A., Lublin, Poland) and *C. albicans* to amphotericin B (NeoSensitabs, Rosco, Denmark) was tested by diffusion method. The experiment was carried out in triplicate.

In order to determine the shape, dimensions, and cell membrane damage of the bac-teria, two representative strains, *S. aureus* ATCC 25923 and *E. coli* ATCC 25922, were select-ed, and an atomic force microscope (AFM) was used. Bacterial strains were cultured in MH broth in the presence of AgChS3 (10 mg/mL) for 18 h at 37 °C. The control was the medium without the addition of the compound.

### 4.5. Adsorption from Aqueous Solution Studies

The adsorption of the dye Acid Red 88 from aqueous solutions on the AgNP–chitosan–silica composites was carried out using a static method. Three series of adsorption systems were prepared by adding 25 mL of adsorbate solution of a fixed concentration (0.45–2.25 mmol/L) in Erlenmeyer flasks containing ~0.05 g of adsorbent. The adsorption process was carried out for two days under specific conditions, i.e., at a temperature of 25 °C and a stirring speed of 120 rpm (Innova 40R model, New Brunswick, NJ, USA). After this time, the equilibrium solutions were filtered and measured spectrophotometrically (UV–Vis spectrophotometer Cary 4000, Varian Inc., Melbourne, VIC, Australia). The amounts of dye adsorbed on the composites at equilibrium were determined using the mass balance equation.

The kinetics of dye adsorption on AgNP–chitosan–silica composites were investigated using the continuous recording technique of absorption spectra [59]. In detail, to a thermostatic glass vessel (25 °C, Thermostat Ecoline RE 207, Lauda, Germany) containing 0.05 g of adsorbent, 200 mL of a 0.076 mmol/L dye solution was added. Samples were automatically withdrawn from the reaction solution into a flow cell at specific time intervals and subjected to spectrophotometric measurements (UV–Vis spectrophotometer Cary 100, Varian Inc., Melbourne, VIC, Australia). The measured sample of the solution was returned to the reaction vessel. The adsorption system was mechanically stirred (120 rpm) throughout the experiment. The changes in absorbance at the peak maximum of the successively recorded absorption spectra were converted into changes in dye concentration as a function of time.

### 4.6. Characterization Methods of Materials

The physicochemical properties of the pure SBA-15 carrier and the modified composites were analyzed by UV–Vis reflectance spectroscopy (UV–Vis DRS). The UV–Vis reflectance spectra of the samples were recorded using a Varian Cary 4000 spectrophotometer. equipped with a diffuse reflectance accessory and an integrating sphere (Varian Inc., Melbourne, VIC, Australia). The materials were placed in a powder cell and the diffuse reflectance spectra were recorded over the wavelength range of 200–800 nm.

The textural properties of the composites were evaluated by low-temperature nitrogen adsorption–desorption isotherms measured at 77 K over the entire range of relative pressures (0 to 950 mmHg) using a sorption analyzer (ASAP 2020, Micromeritics, Norcross, GA, USA). The specific surface area (S_BET_) was determined from the experimental isotherms using the standard BET method. The pore size distribution curves were derived from the adsorption and desorption branches of the isotherms using the Barrett–Joyner–Halenda (BJH) model with cylindrical pores and the Faas correction. The total pore volume (V_t_) was calculated based on the nitrogen adsorbed at p/p_0_ = 0.98. For the values of the external (macropore) surface area, S_ext_, and the primary mesopore volume, V_p_, the α_s_ plot method was used with the macroporous silica gel LiChrospher Si-1000 as a reference non-porous adsorbent [60], while the micropore volume (V_mic_) was estimated using the t-plot method. Prior to analysis, all samples were degassed for 24 h at 90 °C and 1 mmHg in the degassing port of the analyzer.

The crystallinity of the AgChS1–AgChS3 composites was investigated by powder X-ray diffraction (XRD; Empyrean diffractometer (PANalytical, Malvern, United Kingdom) using CuKa radiation (k = 1.54 Å), recording over a 1 h range from 10° to 90°. The instrument was operated at 40 kV with a current of 40 mA. The Ag crystallite size LAg was determined by the Scherrer equation from the full width at half maximum of the X-ray diffraction peaks [61]. The SAXS analysis was conducted using X-ray diffraction (XRD) equipped with a Cu anode X-ray tube and the SAXS/WAXS sample stage in capillary mode. The device was powered by a 4 kW high-voltage X-ray generator with settings of 40 kV and 40 mA. The incident beam path utilized W/Si and a graded elliptical X-ray mirror. The SAXS setup operated with a 2θ range of −0.1 to 4 degrees, a step size of 0.005, and a counting time of 1.76 s, resulting in 821 points per scan. A Cu 0.2 mm beam attenuator was used near primary beam measurements. Data were collected via a PIXcel3D detector and a receiving slit of 0.05 mm active length. The scattering vector, q, was defined as q = 4πsinθ/λ, where 2θ is the scattering angle and λ is the X-ray wavelength (1.5418 Å). Background scattering was assessed using air-scattering measurements with an empty sample holder.

Transmission electron micrographs were obtained from a high-resolution transmission electron microscope Titan G2 60-300 (FEI, Thermo Fisher Scientific, Hillsboro, OR, USA) operating at 200 kV. TEM specimens were prepared by directly drying a drop of a dilute ethanol dispersion solution of the solid sample on the surface of a carbon-coated copper grid. Before measurement, the grid with the sample applied was air-dried. The colloidal silver solution sample was applied directly to the measuring grid and gently air-dried.

Atomic force microscopy (AFM) analysis was performed for the illustration 3D topography on a Bruker–Veeco–Digital Instruments Multi-Mode Atomic Force Microscope (Bruker, Germany). The dynamic mode (tapping) was applied during AFM imaging. The NanoScope Analysis software, version 1.40 delivered from Bruker (Bruker, Germany) was applied for the data treatment.

The surface charge properties were assessed through potentiometric titration of the suspension using a 765 Dosimat Autoburette (Metrohm, Herisau, Switzerland) in conjunction with a PHM240 pH meter (Radiometer, Copenhagen, Denmark). Experiments were conducted at a constant temperature of 25 °C, maintained by an Ecoline RE207 thermostat (Lauda, Germany), with data acquisition managed by the Titr_v3 software (developed by Marczewski and Janusz, Faculty of Chemistry, Maria Curie-Sklodowska University, Lublin, Poland). The specific experimental conditions were as follows: 30 mL of 0.1 mol/L NaCl as the electrolyte, 0.3 mL of 0.5 mol/L HCl to stabilize the initial pH, 0.10 g of each solid sample, and 0.2 mol/L NaOH as the titrant. Measurements were conducted under a nitrogen atmosphere to prevent carbon dioxide contamination.

Elemental analysis for carbon, hydrogen, and nitrogen in the AgNP–chitosan–silica composites was conducted using a Series II CHNS/O Analyzer 2400 (Perkin Elmer, Waltham, MA, USA). The reduction and combustion processes were performed at temperatures of 650 °C and 950 °C, respectively, with 500 mg of each composite sample used for the analysis.

## Figures and Tables

**Figure 1 ijms-25-13548-f001:**
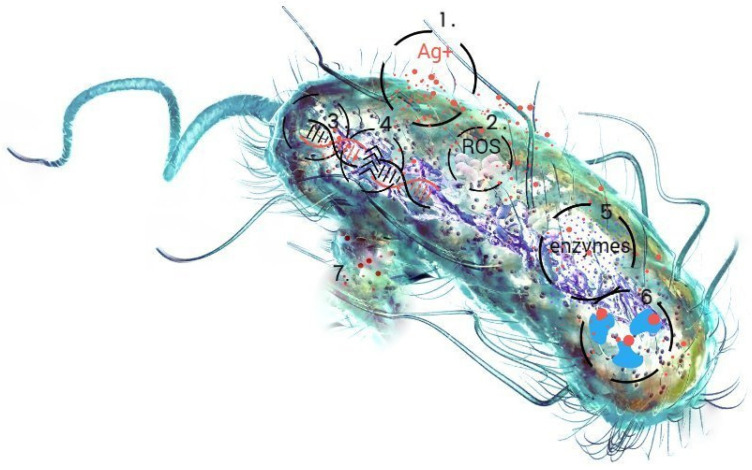
Schematic representation of the action of silver nanoparticles on bacterial cells using the example of the *E. coli* bacteria model.

**Figure 2 ijms-25-13548-f002:**
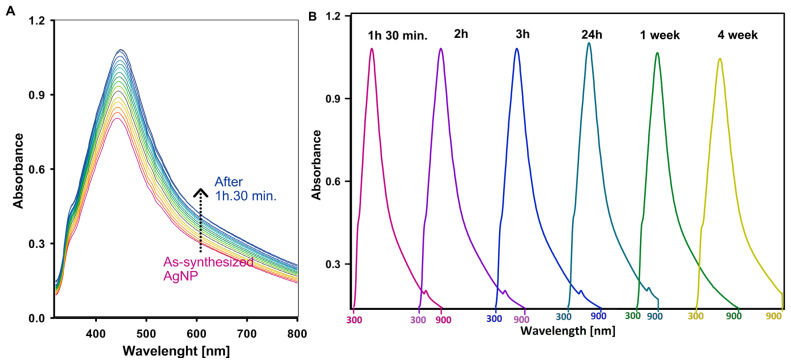
The process of silver ion reduction controlled by UV–Vis technique: (**A**) changes in the concentration of silver nanoparticles in the process of reduction of diamminesilver(I) ions; and (**B**) UV–Vis spectra of silver nanoparticle solutions recorded after specific time intervals from the establishment of equilibrium. Each spectrum in Figure 2B was recorded in the wavelength range of 300 to 900 nm.

**Figure 3 ijms-25-13548-f003:**
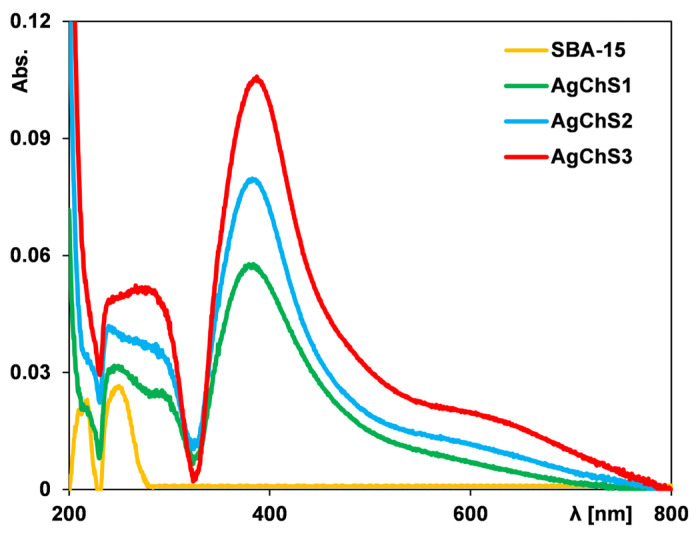
UV–Vis spectra for pure SBA-15 and composites with different contents of silver nanoparticles.

**Figure 4 ijms-25-13548-f004:**
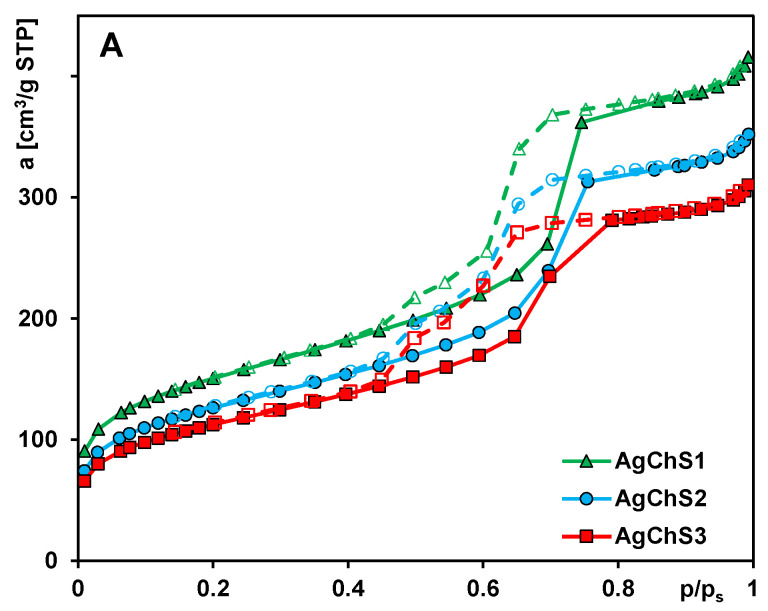
(**A**) Nitrogen adsorption–desorption isotherms at 77 K for the analyzed samples, (**B**) porosity distributions calculated with the BJH theory from the adsorption, and (**C**) desorption branches of the isotherms as a function of pore size (in a differential form dV/dD, where V and D are pore volume and diameter, respectively).

**Figure 5 ijms-25-13548-f005:**
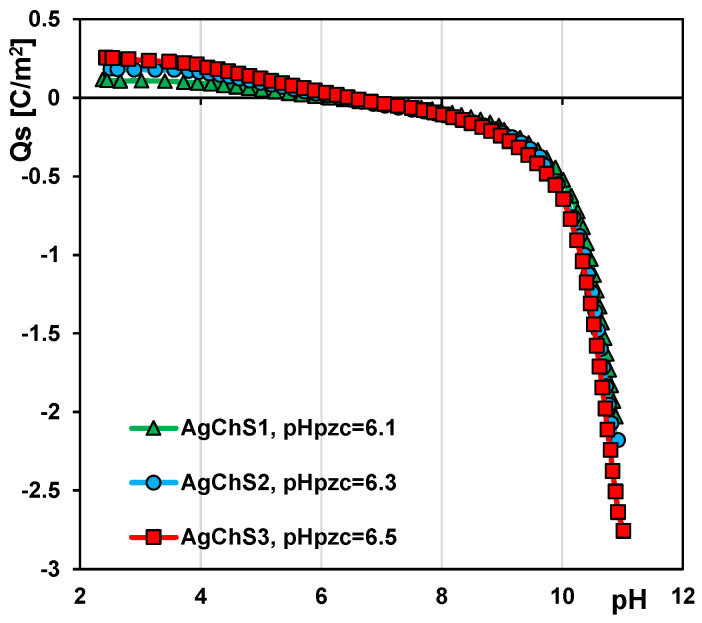
Dependences of surface charge density of the composites AgChS1–AgChS3 on solution pH.

**Figure 6 ijms-25-13548-f006:**
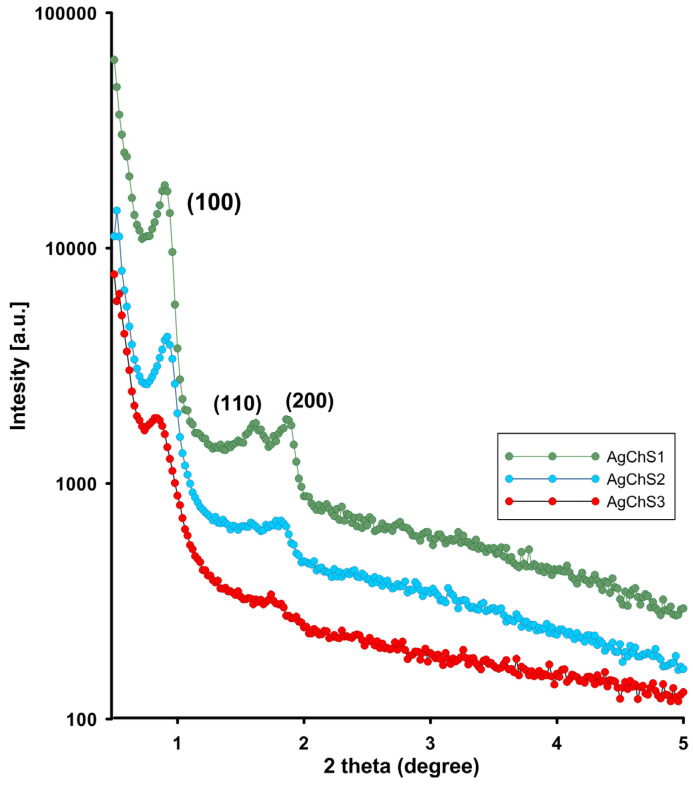
The powder XRD patterns in the range of small diffraction angles for AgChS1–AgChS3 samples.

**Figure 7 ijms-25-13548-f007:**
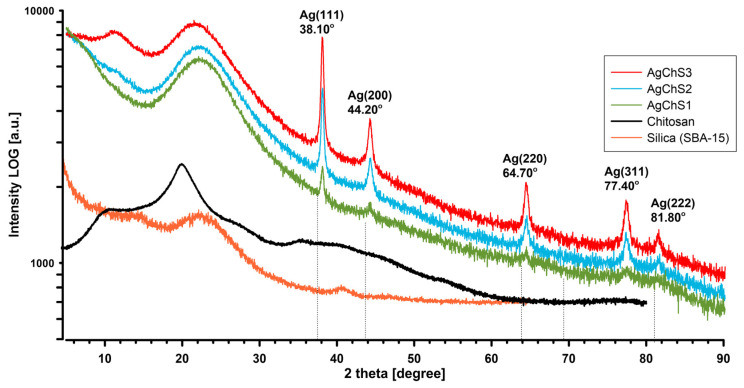
XRD images of the biopolymer–nanosilver composites AgChS1–AgChS3 and experimental comparison curves of chitosan and silica components.

**Figure 8 ijms-25-13548-f008:**
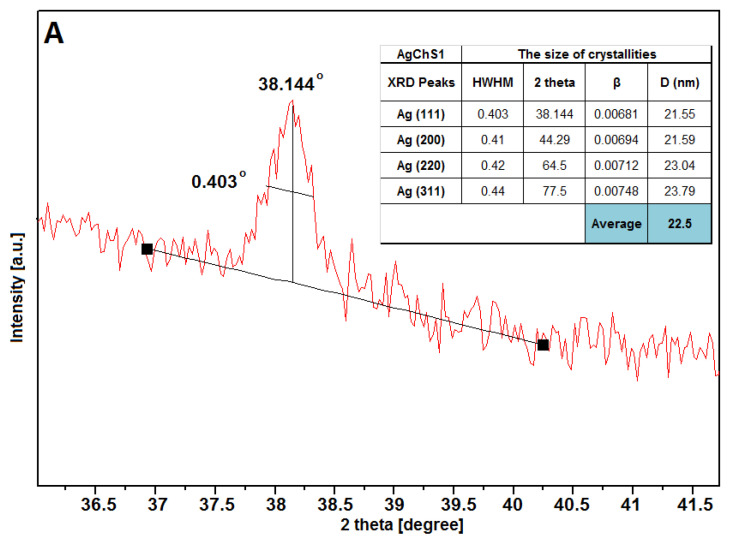
Parameterization of the first XRD Ag(111) signal of the tested materials AgChS1 (**A**), AgChS2 (**B**), and AgChS3 (**C**). Inset tables show calculations of the crystallite size relative to the plane perpendicular to the (111), (200), (220), and (311) directions together with the average silver crystallite size D (nm).

**Figure 9 ijms-25-13548-f009:**
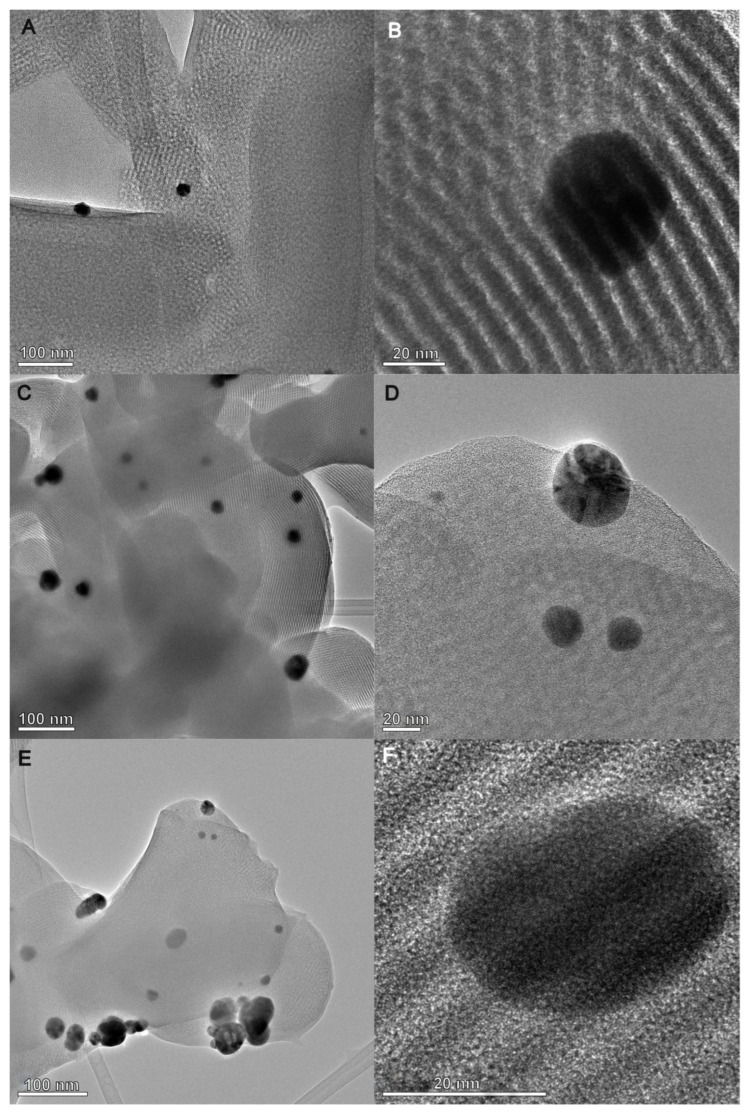
Transmission electron micrographs (TEMs) for the AgNP–chitosan–silica composites AgChS1 (**A**,**B**), AgChS2 (**C**,**D**), and AgChS3 (**E**,**F**).

**Figure 10 ijms-25-13548-f010:**
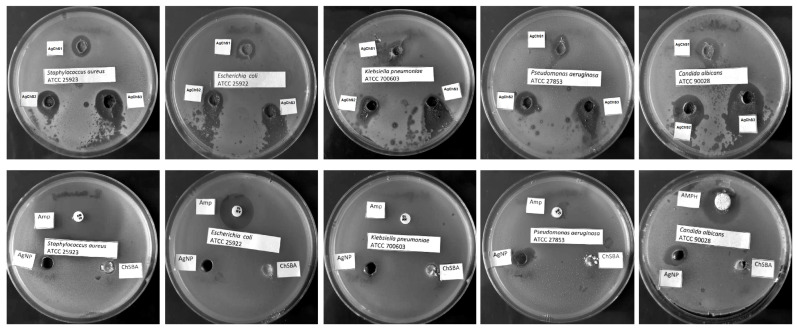
Microbial growth inhibition of: *S. aureus* ATCC 25923, *E. coli* ATCC 25922, *K. pneumoniae* ATCC 700603, *P. aeruginosa* ATCC 27853, and *C. albicans* ATCC 90028 by tested materials: AgChS1–AgChS3, ChSBA, AgNPs, ampicillin (AMP), and amphotericin B (AMPH).

**Figure 11 ijms-25-13548-f011:**
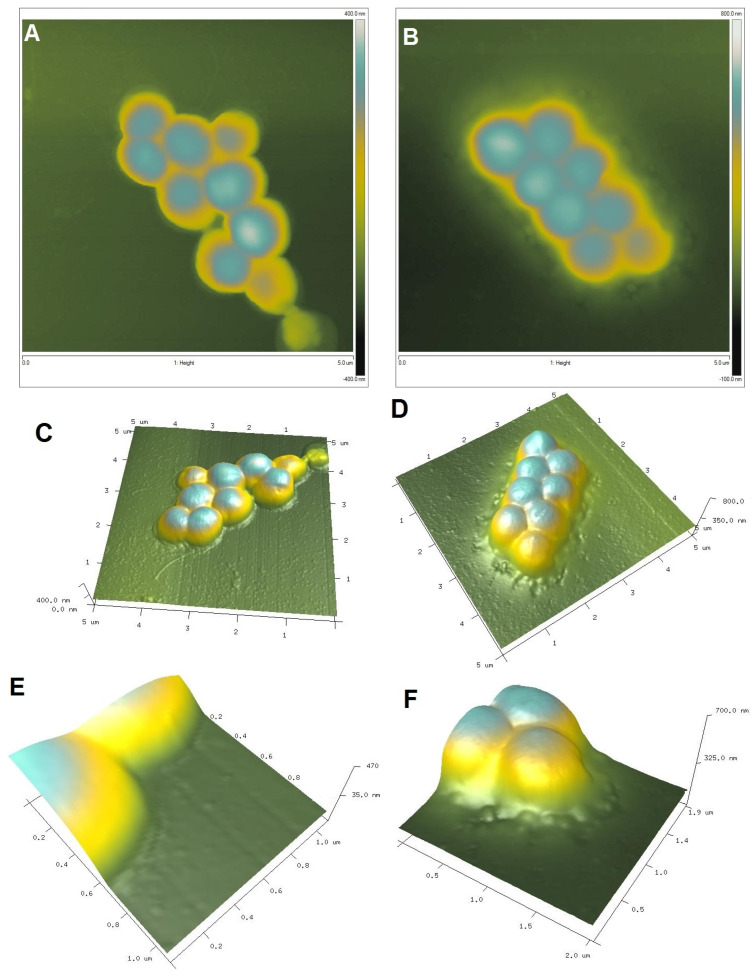
AFM images of Gram-positive *Staphylococcus aureus* untreated (control sample (**A**,**C**,**E**,**G**) and Staphylococcus aureus exposed to AgNP–chitosan–silica composite (AgChS3 (**B**,**D**,**F**,**H**)) (**A**,**B**) topography as 2D view and 3D image of the upper surface of bacterial strains (**C**,**D**), enlargement of the bacterial envelope area of the control bacteria (**E**) and the bacteria exposed to the composite material (**F**), and surface topography of bacteria before (**G**) and after contact with the material (**H**).

**Figure 12 ijms-25-13548-f012:**
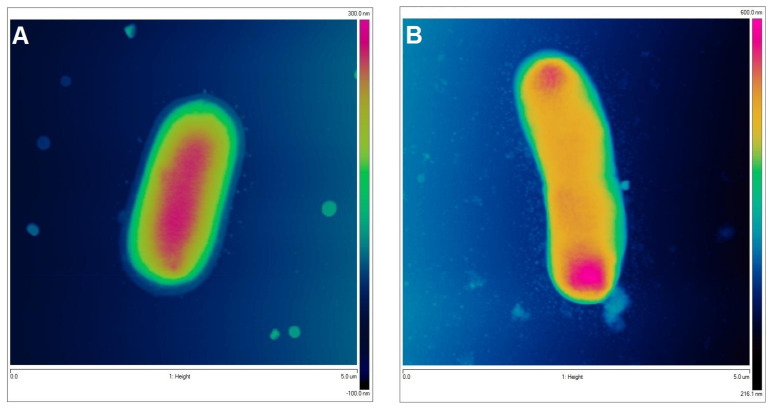
AFM images of Gram-negative *Escherichia coli* untreated (control sample (**A**,**C**,**E**,**G**) and bacteria exposed to AgChS3 (**B**,**D**,**F**,**H**)) (**A**,**B**) topography as 2D view and 3D image of the upper surface of bacterial strains (**C**,**D**), enlargement of the bacterial envelope area of the control bacteria (**E**) and the bacteria exposed to the composite material (**F**), and surface topography of bacteria before (**G**) and after contact with the composite material (**H**).

**Figure 13 ijms-25-13548-f013:**
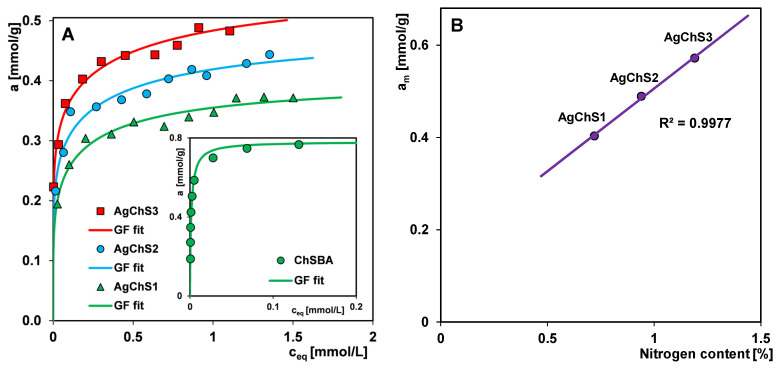
(**A**) Adsorption isotherms of Acid Red 88 on the AgNP–chitosan–silica composites and on the composite ChSBA (as a control material, inset) as a dependence of adsorbed dye amount on equilibrium concentration c_eq_. (**B**) Dependence of adsorption capacity a_m_ on the nitrogen content in the composites.

**Figure 14 ijms-25-13548-f014:**
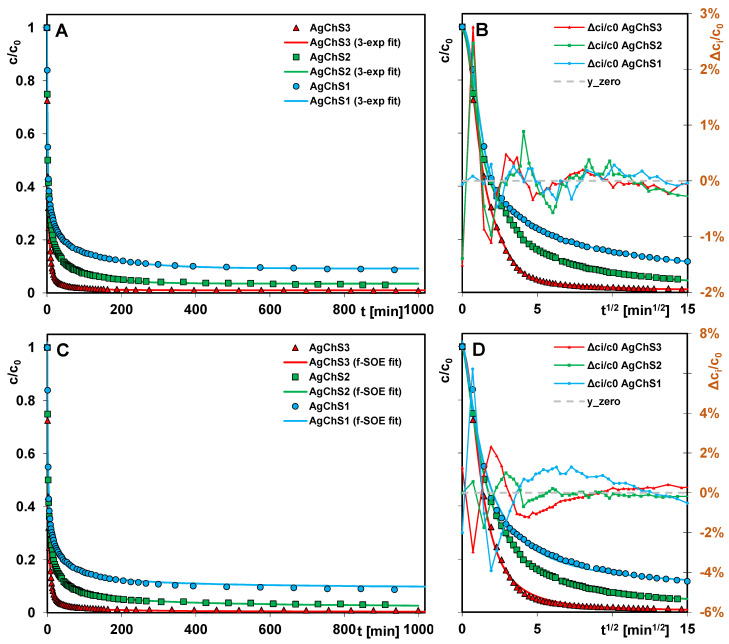
Comparison of Acid Red 88 adsorption kinetics on the AgNP–chitosan–silica composites at coordinates: relative concentration~time (**A**,**C**); relative concentration~square root of time (**B**,**D**). The lines correspond to the fitted m-exponential equation (**A**,**B**) and the fractal-like SOE equation (**C**,**D**).

**Figure 15 ijms-25-13548-f015:**
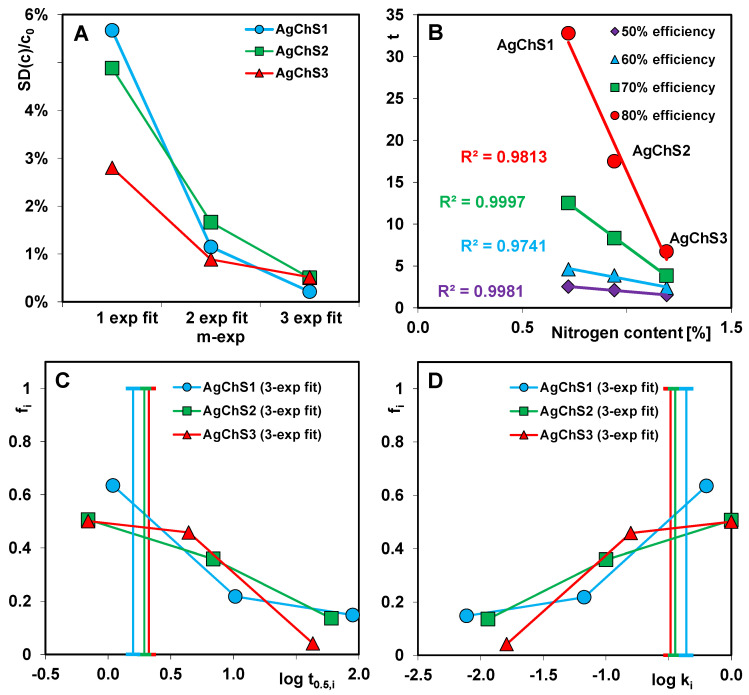
Dependence of standard deviations of relative concentration SD(c)/c_0_ on the number of exponential terms in the multi-exponential equation (**A**); the relationship between adsorption kinetics at fixed values of the process progress and N content in the composites (**B**); and distribution of half-time t_0.5i_ (**C**) and rate coefficient k_i_ (**D**) for dye adsorption on the composites.

**Figure 16 ijms-25-13548-f016:**
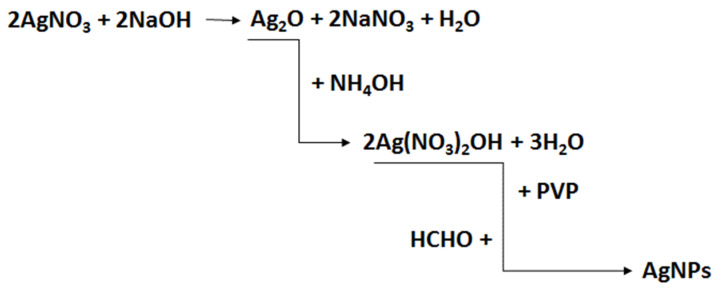
Scheme of the synthesis of stabilized silver nanoparticles solution.

**Table 1 ijms-25-13548-t001:** The parameters of the textural properties obtained from low-temperature nitrogen adsorption data.

Composite	S_BET_ ^a^ [m^2^/g]	S_ext_ ^b^ [m^2^/g]	V_t_ ^c^ [cm^3^/g]	V_p_ ^d^ [cm^3^/g]	V_mic_ ^e^ (t-Plot) [cm^3^/g]	D_BJH ads_ ^f^ [nm]	D_BJH des_ ^g^ [nm]
AgChS1	527	61	0.64	0.52	0.05	6.0	5.4
AgChS2	444	46	0.54	0.45	0.04	5.9	5.2
AgChS3	396	39	0.48	0.40	0.03	5.7	5.0

^a^ BET surface area calculated using experimental points at a relative pressure of (p/p_0_) 0.035–0.31, where p and p_0_ are denoted as the equilibrium and saturation pressure of nitrogen. ^b^ External surface area calculated by α_s_ method. ^c^ Total pore volume calculated by 0.0015468 amount of nitrogen adsorbed at p/p_0_ = 0.98. ^d^ Primary mesopores volume calculated by α_s_ method. ^e^ Pore volume of micropores calculated by t-plot method with fitted statistical thickness in the range of 3.56 to 4.86 Å. ^f^ The average pore diameter estimated based on the BJH theory from adsorption data. ^g^ The average pore diameter is estimated based on the BJH theory from desorption data.

**Table 2 ijms-25-13548-t002:** The diameters of bacterial growth inhibition zones for AgChS1, AgChS2, and AgChS3 materials. Standard deviation (SD) ranging from ±0.1 and ±0.84 (n = 3). ChSBA—chitosan–silica composite, AgNPs—silver nanoparticles, Amp—ampicillin, AMPH—amphotericin B, nd—not determined.

Strain	AgChS1;10 mg	AgChS2;10 mg	AgChS3;10 mg	ChSBA;10 mg	AgNPs;50 μL	Amp;10 μg	AMPH;10 μg
	Growth Inhibition Zone [mm]						
*Staphylococcus aureus* ATCC 25923	12	15	17	0	8	30	nd
*Escherichia coli* ATCC 25922	11	13	14	0	7	20	nd
*Klebsiella pneumoniae* ATCC 700603	10	11	12	0	7	0	nd
*Pseudomonas aeruginosa* ATCC 27853	10	13	14	0	9	0	nd
*Candida albicans* ATCC 90028	14	18	21	0	11	nd	21

**Table 3 ijms-25-13548-t003:** The parameters of the Generalized Freundlich equation (GF) for the Acid Red 88 adsorption on the composites.

Composite	a_m_ [mmol/g]	m	n	Log K	R^2^
AgChS1	0.40	0.20	1	0.04	0.96
AgChS2	0.49	0.17	1	−0.19	0.95
AgChS3	0.57	0.14	1	−0.34	0.98

**Table 4 ijms-25-13548-t004:** Amounts of reagents used for the synthesis of AgNP–chitosan–silica composites and their elemental composition.

Code	Reagents for AgNP–Silica Phase			Reagents for Composite		Elemental Composition		
	AgNP [mL]	Water [mL]	Silica [g]	AgNP–Silica Phase [g]	Chitosan [g]	C [%]	H [%]	N [%]
AgChS1	15	15	2	2	0.2	4.2	2.0	0.72
AgChS2	23	7	2	2	0.3	5.3	2.3	0.94
AgChS3	30	-	2	2	0.4	6.8	2.4	1.19

**Table 5 ijms-25-13548-t005:** Comparison of the kinetic parameters determined from the multi-exponential equation and the fractal-like SOE equation.

Composite	Fit	f_2_/p	Log k	t_0.5_ [min]	u_eq_	SD(c/c_0_) [%]	1-R^2^
AgChS1	3-exp	-	−0.49	2.12	0.91	0.21	1.30 × 10^−4^
	f-SOE	1/0.70	−0.34	2.18	0.92	1.05	4.62 × 10^−3^
AgChS2	3-exp	-	−0.45	1.95	0.96	0.50	7.75 × 10^−4^
	f-SOE	1/0.71	−0.35	2.23	0.98	0.39	4.95 × 10^−4^
AgChS3	3-exp	-	−0.36	1.58	0.99	0.52	8.00 × 10^−4^
	f-SOE	1/0.80	−0.21	1.62	1	0.83	2.25 × 10^−3^

**Table 6 ijms-25-13548-t006:** Comparison of times needed to obtain the set decolorization efficiency using the composites.

Efficiency	Time [min]		
	AgChS1	AgChS2	AgChS3
50%	2.55	2.12	1.55
60%	4.61	3.89	2.40
70%	12.52	8.33	3.85
80%	32.79	17.50	6.75

## Data Availability

The data presented in this study are available upon request from the corresponding author.

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
