# Peer review of "Nanosilver–Biopolymer–Silica Composites: Preparation, and Structural and Adsorption Analysis with Evaluation of Antimicrobial Properties"

_ijms, 2024, doi:10.3390/ijms252413548_

Round 1
Reviewer 1 Report
Comments and Suggestions for Authors
Hello
The work is interesting, but the exposition is cumbersome and incoherent.
Most of the parameters, functions and quantities used in the graphs are not explained, defined. Only in equations have the authors defined some of the variables, but not all, and especially their importance is not specified.
Regarding the figures
Figure 2B does not have the y-axis marked
Fig 3 – lack of control line for Ag
Missing explanation of parameters on axes fig 4.
The legend in fig. 7 does not agree with the graph lines
Figure 8 is not very clear, the writing is very small
Figure 11 B and 13 a representation from 3 points (inconclusive)
An important element missing from the adsorption study is the comparison with a control: a chitosan or chitosan-silica matrix, to emphasize the importance of Ag NPs doping
Numerous parameters that were represented, and which were not explained in the adsorption experiment (nitrogen content, concentration at equilibrium, a, am, fi, efficiency, etc.). And on the other hand, other parameters are presented in the equations and are not discussed/represented.
The conclusions support the AgChS3 sample for the best antimicrobial and adsorptive activity, but for the purpose of depollution, did the authors take into account a possible release of AgNPs in water depolluted by simulated dye?
Reviewer 2 Report
Comments and Suggestions for Authors
The authors have developed SBA-15 silica composites modified with AgNPs and chitosan as antimicrobial agents and adsorption phases. Additionally, AgChS have exhibited potential for water purification, particularly with a high chitosan content composite showing excellent adsorption capacity for Acid Red 88 dye, highlighting their dual functionality as both antimicrobial and environmentally cleansing materials. Overall, this work has demonstrated the successful preparation of AgChS in antimicrobial application and adsorption. I would like to recommend this work for the publication in IJMS after revision. Herein, I provide some suggestions for this work.
1. This paper would be more impressive if the authors could provide the SEM images of bacteria after culture with AgChS. These data could provide the demonstration of damage of bacteria by AgChS.
2. Please provide the TEM image of AgNPs in the supporting information.
3. To demonstrate the composites of AgChS1, AgChS2 and AgChS3, elemental mapping is recommended to provide in the supporting information.
4. For the introduction “Thus, the studies on the development of new effective and selective adsorbents…”, some references could be added to enhance the importance.
https://doi.org/10.3390/nano14080695
Round 2
Reviewer 1 Report
Comments and Suggestions for Authors
I ask the authors to pay attention to the way microorganisms are written.
And also to improve the resolution of the figures in SI (especially in figure 2SI
Reviewer 2 Report
Comments and Suggestions for Authors
The authors have replied the comments from the reviewers. The revision have been made based on the comments. The revision now is suitable for the publication as its current form in IJMS.
